# Learnable Embedding Space for Efficient Neural Architecture Compression

**Shengcao Cao***
School of EECS
Peking University
Beijing, 100871, China
caoshengcao@pku.edu.cn

**Xiaofang Wang*** **& Kris M. Kitani**
The Robotics Institute
Carnegie Mellon University
Pittsburgh, PA 15213, USA
{xiaofan2,kkitani}@cs.cmu.edu

## Abstract

We propose a method to incrementally learn an embedding space over the domain of network architectures, to enable the careful selection of architectures for evaluation during compressed architecture search. Given a teacher network, we search for a compressed network architecture by using Bayesian Optimization (BO) with a kernel function defined over our proposed embedding space to select architectures for evaluation. We demonstrate that our search algorithm can significantly outperform various baseline methods, such as random search and reinforcement learning (Ashok et al., 2018). The compressed architectures found by our method are also better than the state-of-the-art manually-designed compact architecture ShuffleNet (Zhang et al., 2018). We also demonstrate that the learned embedding space can be transferred to new settings for architecture search, such as a larger teacher network or a teacher network in a different architecture family, without any training.

## 1 Introduction

In many application domains, it is common practice to make use of well-known deep network architectures (*e.g.,* VGG (Simonyan & Zisserman, 2014), GoogleNet (Szegedy et al., 2015), ResNet (He et al., 2016)) and to adapt them to a new task without optimizing the architecture for that task. While this process of transfer learning is surprisingly successful, it often results in over-sized networks which have many redundant or unused parameters. Inefficient network architectures can waste computational resources and over-sized networks can prevent them from being used on embedded systems. There is a pressing need to develop algorithms that can take large networks with high accuracy as input and compress their size while maintaining similar performance. In this paper, we focus on the task of compressed architecture search – the automatic discovery of compressed network architectures based on a given large network.

One significant bottleneck of compressed architecture search is the need to repeatedly evaluate different compressed network architectures, as each evaluation is extremely costly (*e.g.,* back-propagation to learn the parameters of a single deep network can take several days on a single GPU). This means that any efficient search algorithm must be judicious when selecting architectures to evaluate. Learning a good embedding space over the domain of compressed network architectures is important because it can be used to define a distribution on the architecture space that can be used to generate a priority ordering of architectures for evaluation. To enable the careful selection of architectures for evaluation, we propose a method to incrementally learn an embedding space over the domain of network architectures.

In the network compression paradigm, we are given a teacher network and we aim to search for a compressed network architecture, a student network that contains as few parameters as possible while maintaining similar performance to the teacher network. We address the task of compressed architecture search by using Bayesian Optimization (BO) with a kernel function defined over our proposed embedding space to select architectures for evaluation. As modern neural architectures can

---

*indicates equal contribution.

have multiple layers, multiple branches and multiple skip connections, defining an embedding space over all architectures is non-trivial. In this work, we propose a method for mapping a diverse range of discrete architectures to a continuous embedding space through the use of recurrent neural networks. The learned embedding space allows us to perform BO to efficiently search for compressed student architectures that are also expected to have high accuracy.

We demonstrate that our search algorithm can significantly outperform various baseline methods, such as random search and reinforcement learning (Ashok et al., 2018). For example, our search algorithm can compress VGG-19 (Simonyan & Zisserman, 2014) by $8\times$ on CIFAR-100 (Krizhevsky & Hinton, 2009) while maintaining accuracy on par with the teacher network. The automatically found compressed architectures can also achieve higher accuracy than the state-of-the-art manually-designed compact architecture ShuffleNet (Zhang et al., 2018) with a similar size. We also demonstrate that the learned embedding space can be transferred to new settings for architecture search, such as a larger teacher network or a teacher network in a different architecture family, without any training.

**Contributions**: (1) We propose a novel method to incrementally learn an embedding space over the domain of network architectures. Based on the learnable embedding space, we present a framework of searching for compressed network architectures with BO. The learned embedding provides a feature space over which the kernel function of BO is defined. (2) We propose a set of architecture operators for generating architectures for search. Operators for modifying the teacher network are: layer removal, layer shrinkage and skip connection addition. (3) We propose a multiple kernel strategy to prevent the premature convergence of the search and encourage the search algorithm to explore more diverse architectures during the search process.

## 2 RELATED WORK

**Computationally Efficient Architecture**: There has been great progress in designing computationally efficient network architectures. Representative examples include SqueezeNet (Iandola et al., 2016), MobileNet (Howard et al., 2017), MobileNetV2 (Sandler et al., 2018), CondenseNet (Huang et al., 2018) and ShuffleNet (Zhang et al., 2018). Different from them, we aim to develop an algorithm that can automatically search for an efficient network architecture with minimal human efforts involved in the architecture design.

**Neural Architecture Search (NAS)**: NAS has recently been an active research topic (Zoph & Le, 2016; Zoph et al., 2017; Real et al., 2018; Pham et al., 2018; Liu et al., 2017a;b; 2018; Luo et al., 2018). Some existing works in NAS are focused on searching for architectures that not only can achieve high performance but also respect some resource or computation constraints (Ashok et al., 2018; Tan et al., 2018; Zhou et al., 2018; Dong et al., 2018; Hsu et al., 2018; Elsken et al., 2018a). NAO (Luo et al., 2018) and our work share the idea of mapping network architectures into a latent continuous embedding space. But NAO and our work have fundamentally different motivations, which further lead to different architecture search frameworks. NAO maps network architectures to a continuous space such that they can perform gradient based optimization to find better architectures. However, our motivation for learning the embedding space is to find a principled way to define a kernel function between architectures with complex skip connections and multiple branches.

Our work is also closely related to N2N (Ashok et al., 2018), which searches for a compressed architecture based on a given teacher network using reinforcement learning. Our search algorithm is developed based on Bayesian Optimization (BO), which is different from N2N and many other existing works. We will compare our approach to other BO based NAS methods in the next paragraph. Readers can refer to Elsken et al. (2018b) for a more complete literature review of NAS.

**Bayesian Optimization (BO)**: BO is a popular method for hyper-parameter optimization in machine learning. BO has been used to tune the number of layers and the size of hidden layers (Bergstra et al., 2011; Swersky et al., 2014), the width of a network (Snoek et al., 2012) or the size of the filter bank (Bergstra et al., 2013), along with other hyper-parameters, such as the learning rate, number of iterations. Mendoza et al. (2016), Jenatton et al. (2017) and Zela et al. (2018) also fall into this category. Our work is also related to Hernández-Lobato et al. (2016), which presents a Bayesian method for identifying the Pareto set of multi-objective optimization problems and applies the method to searching for a fast and accurate neural network.

However, most existing works on BO for NAS only show results on tuning network architectures where the connections between network layers are fixed, i.e., most of them do not optimize how the layers are connected to each other. Kandasamy et al. (2018) proposes a distance metric OTMANN to compare network architectures with complex skip connections and branch layers, based on which NASBOT is developed, a BO based NAS framework, which can tune how the layers are connected. Although the OTMANN distance is designed with thoughtful choices, it is defined based on some empirically identified factors that can influence the performance of a network, rather than the actual performance of networks. Different from OTMANN, the distance metric (or the embedding) for network architectures in our algorithm is automatically learned according to the actual performance of network architectures instead of manually designed.

Our work can also be viewed as carrying out optimization in the latent space of a high dimensional and structured space, which shares a similar idea with previous literature (Lu et al., 2018; Gómez-Bombarelli et al., 2018). For example, Lu et al. (2018) presents a new variational auto-encoder to map kernel combinations produced by a context-free grammar into a continuous latent space.

**Deep Kernel Learning**: Our work is also related to recent works on deep kernel learning (Wilson et al., 2016a;b). They aim to learn more expressive kernels by representing the kernel function as a neural network to incorporate the expressive power of deep networks. The follow-up work (Al-Shedivat et al., 2017) extends the kernel representation to recurrent networks to model sequential data. Our work shares a similar motivation with them and tries to learn a kernel function for the neural architecture domain by leveraging the expressive power of deep networks.

## 3 APPROACH

In this work, we focus on searching for a compressed network architecture based on a given teacher network and our goal is to find a network architecture which contains as few parameters as possible but still can obtain a similar performance to the teacher network. Formally, we aim to solve the following optimization problem:

$$x^* = \arg\max_{x \in \mathcal{X}} f(x), \tag{1}$$

where $\mathcal{X}$ denotes the domain of neural architectures and the function $f(x) : \mathcal{X} \mapsto \mathbb{R}$ evaluates how well the architecture $x$ meets our requirement. We adopt the reward function design in N2N (Ashok et al., 2018) for the function $f$, which is defined based on the compression ratio of the architecture $x$ and its validation performance after being trained on the training set. More details about the exact form of $f$ are given in Appendix 6.1 due to space constraints.

As evaluating the value of $f(x)$ for a specific architecture $x$ is extremely costly, the algorithm must judiciously select the architectures to evaluate. To enable the careful selection of architectures for evaluation, we propose a method to incrementally learn an embedding space over the domain of network architecture that can be used to generate a priority ordering of architectures for evaluation. In particular, we develop the search algorithm based on BO with a kernel function defined over our proposed embedding space. In the following text, we will first introduce the sketch of the BO algorithm and then explain how the proposed embedding space is used in the loop of BO.

We adopt the Gaussian process (GP) based BO algorithms to maximize the function $f(x)$, which is one of the most popular algorithms in BO. A GP prior is placed on the function $f$, parameterized by a mean function $\mu(\cdot) : \mathcal{X} \mapsto \mathcal{R}$ and a covariance function or kernel $k(\cdot, \cdot) : \mathcal{X} \times \mathcal{X} \mapsto \mathcal{R}$. To search for the solution, we start from an arbitrarily selected architecture $x_1$. At step $t$, we evaluate the architecture $x_t$, i.e., obtaining the value of $f(x_t)$. Using the $t$ evaluated architectures up to now, we compute the posterior distribution on the function $f$:

$$p\left(f(x) \mid f(x_{1:t})\right) \sim \mathcal{N}(\mu_t(x), \sigma_t^2(x)), \tag{2}$$

where $f(x_{1:t}) = [f(x_1), \ldots, f(x_t)]$ and $\mu_t(x)$ and $\sigma_t^2(x)$ can be computed analytically based on the GP prior (Williams & Rasmussen, 2006). We then use the posterior distribution to decide the next architecture to evaluate. In particular, we obtain $x_{t+1}$ by maximizing the expected improvement acquisition function $\mathrm{EI}_t(x) : \mathcal{X} \mapsto \mathbb{R}$, i.e., $x_{t+1} = \arg\max_{x \in \mathcal{X}} \mathrm{EI}_t(x)$. The expected improvement function $\mathrm{EI}_t(x)$ (Mockus & Mockus, 1991) measures the expected improvement over the current maximum value according to the posterior distribution:

$$\mathrm{EI}_t(x) = \mathbb{E}_t[\max(0, f(x) - f_t^*)], \tag{3}$$

where $\mathbb{E}_t$ indicates the expectation is taken with respect to the posterior distribution at step $t$ $p\left(f(x) \mid f(x_{1:t})\right)$ and $f_t^*$ is the maximum value among $\{f(x_1), \ldots, f(x_t)\}$. Once obtaining $x_{t+1}$, we repeat the above described process until we reach the maximum number of steps. Finally, we return the best evaluated architecture as the solution.

The main difficulty in realizing the above optimization procedure is the design of the kernel function $k(\cdot, \cdot)$ for $\mathcal{X}$ and the maximization of the acquisition function $\mathrm{EI}_t(x)$ over $\mathcal{X}$, since the neural architecture domain $\mathcal{X}$ is discrete and highly complex. To overcome these difficulties, we propose to learn an embedding space for the neural architecture domain and define the kernel function based on the learned embedding space. We also propose a search space, a subset of the neural architecture domain, over which maximizing the acquisition function is feasible and sufficient.

## 3.1 Learnable Embedding Space and Kernel Function

The kernel function, which measures the similarity between network architectures, is fundamental for selecting the architectures to evaluate during the search process. As modern neural architectures can have multiple layers, multiple branches and multiple skip connections, comparing two architectures is non-trivial. Therefore, we propose to map a diverse range of discrete architectures to a continuous embedding space through the use of recurrent neural networks and then define the kernel function based on the learned embedding space.

We use $h(\cdot; \theta)$ to denote the architecture embedding function that generates an embedding for a network architecture according to its configuration parameters. $\theta$ represents the weights to be learned in the architecture embedding function. With $h(\cdot; \theta)$, we define the kernel function $k(x, x'; \theta)$ based on the RBF kernel:

$$k(x, x'; \theta) = \exp\left(-\frac{||h(x; \theta) - h(x'; \theta)||^2}{2\sigma^2}\right), \tag{4}$$

where $\sigma$ is a hyper-parameter. $h(\cdot; \theta)$ represents the proposed learnable embedding space and $k(x, x'; \theta)$ is the learnable kernel function. They are parameterized by the same weights $\theta$. In the following text, we will first introduce the architecture embedding function $h(\cdot; \theta)$ and then describe how we learn the weights $\theta$ during the search process.

The architecture embedding function needs to be flexible enough to handle a diverse range of architectures that may have multiple layers, multiple branches and multiple skip connections. Therefore we adopt a Bidirectional LSTM to represent the architecture embedding function, motivated by the layer removal policy network in N2N (Ashok et al., 2018). The input to the Bi-LSTM is the configuration information of each layer in the network, including the layer type, how this layer connects to other layers, and other attributes. After passing the configuration of each layer to the Bi-LSTM, we gather all the hidden states, apply average pooling to these hidden states and then apply $L_2$ normalization to the pooled vector to obtain the architecture embedding.

We would like to emphasize that our representation for layer configuration encodes the skip connections between layers. Skip connections have been proven effective in both human designed network architectures, such as ResNet (He et al., 2016) and DenseNet (Huang et al., 2017), and automatically found network architectures (Zoph & Le, 2016). N2N only supports the kind of skip connections used in ResNet (He et al., 2016) and does not generalize to more complex connections between layers, where our representation is still applicable. We give the details about our representation for layer configuration in Appendix 6.2.

The weights of the Bi-LSTM $\theta$, are learned during the search process. The weights $\theta$ determine the architecture embedding function $h(\cdot; \theta)$ and the kernel $k(\cdot, \cdot; \theta)$. Further, $\theta$ controls the GP prior and the posterior distribution of the function value conditioned on the observed data points. The posterior distribution guides the search process and is essential to the performance of our search algorithm. Our goal is to learn a $\theta$ such that the function $f$ is consistent with the GP prior, which will result in a posterior distribution that accurately characterizes the statistical structure of the function $f$.

Let $D$ denote the set of evaluated architectures. In step $t$, $D = \{x_1, \ldots, x_t\}$. For any architecture $x_i$ in $D$, we can compute $p\left(f(x_i) \mid f(D \setminus x_i); \theta\right)$ based on the GP prior, where $\setminus$ refers to the set difference operation, $f(x_i)$ is the value obtained by evaluating the architecture $x_i$ and $f(D \setminus x_i) = [f(x_1), \ldots, f(x_{i-1}), f(x_{i+1}) \ldots, f(x_t)]$. $p\left(f(x_i) \mid f(D \setminus x_i); \theta\right)$ is the posterior probability of $f(x_i)$ conditioned on the other evaluated architectures in $D$. The higher the value of

$p\left(f(x_i) \mid f(D \setminus x_i); \theta\right)$ is, the more accurately the posterior distribution characterizes the statistical structure of the function $f$ and the more the function $f$ is consistent with the GP prior. Therefore, we learn $\theta$ by minimizing the negative log posterior probability:

$$L(\theta) = -\frac{1}{|D|} \sum_{i:x_i \in D} \log p\left(f(x_i) \mid f(D \setminus x_i); \theta\right). \tag{5}$$

$p\left(f(x_i) \mid f(D \setminus x_i); \theta\right)$ is a Gaussian distribution and its mean and covariance matrix can be computed analytically based on $k(\cdot, \cdot; \theta)$. Thus $L$ is differentiable with respect to $\theta$ and we can learn the weights $\theta$ using backpropagation.

## 3.2 Acquisition Function and Search Space

In each optimization step, we obtain the next architecture to evaluate by maximizing the acquisition function $\text{EI}_t(\cdot)$ over the neural architecture domain $\mathcal{X}$. On one hand, maximizing $\text{EI}_t(\cdot)$ over all the network architectures in $\mathcal{X}$ is unnecessary. Since our goal is to search for a compressed architecture based on the given teacher network, we only need to consider those architectures that are smaller than the teacher network. On the other hand, maximizing $\text{EI}_t(\cdot)$ over $\mathcal{X}$ is non-trivial. Gradient-based optimization algorithms cannot be directly applied to optimize $\text{EI}_t(\cdot)$ as $\mathcal{X}$ is discrete. Also, exhaustive exploration of the whole domain is infeasible. This calls for a search space that covers the compressed architectures of our interest and easy to explore. Motivated by N2N (Ashok et al., 2018), we propose a search space for maximizing the acquisition function, which is constrained by the teacher network, and provides a practical method to explore the search space.

We define the search space based on the teacher network. The search space is constructed by all the architectures that can be obtained by manipulating the teacher network with the following three operations: (1) layer removal, (2) layer shrinkage and (3) adding skip connections.

**Layer removal and shrinkage**: The two operations ensure that we only consider architectures that are smaller than the given big network. Layer removal refers to removing one or more layers from the network. Layer shrinkage refers to shrinking the size of layers, in particular, the number of filters in convolutional layers, as we focus on convolutional networks in this work. Different from N2N, we do not consider shrinking the kernel size, padding or other configurable variables and we find that only shrinking the number of filters already yields satisfactory performance.

**Adding skip connections**: The operation of adding skip connections is employed to increase the network complexity. N2N (Ashok et al., 2018), which uses reinforcement learning to search for compressed network architectures, does not support forming skip connections in their action space. We believe when searching for compressed architectures, adding skip connections to the compressed network is crucial for it to achieve similar performance to the teacher network and we will show ablation study results to verify this.

The way we define the search space naturally allows us to explore it by sampling the operations to manipulate the architecture of the teacher network. To optimize the acquisition function over the search space, we randomly sample architectures in the search space by randomly sampling the operations. We then evaluate $\text{EI}_t(\cdot)$ over the sampled architectures and return the best one as the solution. We also have tried using evolutionary algorithm to maximize $\text{EI}_t(\cdot)$ but it yields similar results with random sampling. So for the sake of simplicity, we use random sampling to maximize $\text{EI}_t(\cdot)$. We attribute the good performance of random sampling to the thoughtful design of the operations to manipulate the teacher network architecture. These operations already favor the compressed architectures of our interest.

## 3.3 Multiple Kernel Strategy

We implement the search algorithm with the proposed learnable kernel function but notice that the highest function value among evaluated architectures stops increasing after a few steps. We conjecture this is due to that the learned kernel is overfitted to the training samples since we only evaluate hundreds of architectures in the whole search process. An overfitted kernel may bias the following sampled architectures for evaluation.

To encourage the search algorithm to explore more diverse architectures, we propose a multiple kernel strategy, motivated by the bagging algorithm, which is usually employed to avoid overfitting. In

---

**Algorithm 1** Neural Architecture Search with Bayesian Optimization

---

**Input**: Number of steps $T$. Number of kernels $K$. Teacher network $x_{\text{teacher}}$.
Randomly sample $K$ architectures $x_1^1, \ldots, x_1^K$ from the search space defined based on $x_{\text{teacher}}$.
Initialize the set of evaluated architectures $D = \emptyset$.
**for** $t = 1, \ldots, T$ **do**
    Evaluate the $K$ architectures $x_t^1, \ldots, x_t^K$.
    $D = D \cup \{x_t^1, \ldots, x_t^K\}$.
    **for** $k = 1, \ldots, K$ **do**
        Randomly initialize the weights of kernel $k$, denoted as $\theta^k$.
        Randomly sample a subset of $D$, denoted as $D^k$.
        Learn $\theta^k$ on $D^k$ using the objective function in Eq. 5.
        Compute the posterior distribution conditioned on the architectures in $D_k$ with kernel $k$.
        Maximize the acquisition function and denote the solution as $x_{t+1}^k$.
    **end for**
**end for**
Return the best architecture in $D$ as the solution.

---

bagging, instead of training one single model on the whole dataset, multiple models are trained on different subsets of the whole dataset. Likewise, in each step of the search process, we train multiple kernel functions on uniformly sampled subsets of $D$, the set of all the available evaluated architectures. Technically, learning multiple kernels refers to learning multiple architecture embedding spaces, *i.e.*, multiple sets of weights $\theta$. After training the kernels, each kernel is used separately to compute one posterior distribution and determine one architecture to evaluate in the next step. That is to say, if we train $K$ kernels in the current step, we will obtain $K$ architectures to evaluate in the next step. The proposed multiple kernel strategy encourages the search process to explore more diverse architectures and can help find better architectures than training one single kernel only.

When training kernels, we randomly initialize their weights and learn the weights from the scratch on subsets of evaluated architectures. We do not learn the weights of the kernel based on the weights learned in the last step, *i.e.*, fine-tuning the Bi-LSTM from the last step. The training of the Bi-LSTM is fast since we usually only evaluate hundreds of architectures during the whole search process. A formal sketch of our search algorithm in shown Algorithm 1.

## 4 EXPERIMENTS

We first extensively evaluate our algorithm with different teacher architectures and datasets. We then compare the automatically found compressed architectures to the state-of-the-art manually-designed compact architecture, ShuffleNet (Zhang et al., 2018). We also evaluate the transfer performance of the learned embedding space and kernel. We perform ablation study to understand how the number of kernels $K$ and other design choices in our search algorithm influence the performance. Due to space constraints, the ablation study is included in Appendix 6.3.

### 4.1 COMPRESSION EXPERIMENTS

We use two datasets: CIFAR-10 and CIFAR-100 (Krizhevsky & Hinton, 2009). CIFAR-10 contains 60K images in 10 classes, with 6K images per class. CIFAR-100 also contains 60K images but in 100 classes, with 600 images per class. Both CIFAR-10 and CIFAR-100 are divided into a training set with 50K images and a test set with 10K images. We sample 5K images from the training set as the validation set. We provide results on four architectures as the teacher network: VGG-19 (Simonyan & Zisserman, 2014), ResNet-18, ResNet-34 (He et al., 2016) and ShuffleNet (Zhang et al., 2018).

We consider two baselines algorithms for comparison: random search (RS) and a reinforcement learning based approach, N2N (Ashok et al., 2018). Here we use RS to directly maximize the compression objective $f(x)$. To be more specific, RS randomly samples architectures in the search space, then evaluates all of them and returns the best architecture as the optimal solution. In the following experiments, RS evaluates 160 architectures. For our proposed method, we run 20 architecture search steps, where each step generates $K = 8$ architectures for evaluation based on the the

Table 1: Summary of Compression Results.

| CIFAR-100 | | Accuracy | #Params | Ratio | Times | $f(x)$ |
|---|---|---|---|---|---|---|
| VGG-19 | Teacher | 73.71% | 20.09M | - | - | - |
| | Random Search | 68.17% | 2.83M | 0.8593 | 8.04× | 0.9046 |
| | | ±1.28% | ±1.05M | ±0.0525 | ±3.78× | ±0.0074 |
| | Ours | 71.41% | 2.61M | 0.8699 | 7.99× | **0.9518** |
| | | ±0.75% | ±0.61M | ±0.0306 | ±1.99× | **±0.0158** |
| ResNet-18 | Teacher | 78.68% | 11.22M | - | - | - |
| | Random Search | 69.86% | 1.26M | 0.8878 | 10.10× | 0.8752 |
| | | ±1.90% | ±0.54M | ±0.0477 | ±4.33× | ±0.0137 |
| | N2N | 68.01% | 2.42M | 0.7845 | 4.64× | 0.8242 |
| | Ours | 73.83% | 1.87M | 0.8335 | 6.01× | **0.9123** |
| | | ±1.11% | ±0.08M | ±0.0073 | ±0.26× | **±0.0151** |
| ResNet-34 | Teacher | 78.71% | 21.33M | - | - | - |
| | Random Search | 72.33% | 3.61M | 0.8308 | 5.95× | 0.8924 |
| | | ±1.53% | ±0.35M | ±0.0162 | ±0.60× | ±0.0154 |
| | N2N - removal | 70.11% | 4.25M | 0.8008 | 5.02× | 0.8554 |
| | Ours - removal | 74.05% | 3.18M | 0.8508 | 6.88× | 0.9192 |
| | | ±0.52% | ±0.65M | ±0.0307 | ±1.31× | ±0.0033 |
| | Ours | 73.68% | 2.36M | 0.8895 | 9.08× | **0.9246** |
| | | ±0.57% | ±0.15M | ±0.0069 | ±0.59× | **±0.0076** |
| ShuffleNet | Teacher | 71.14% | 1.06M | - | - | - |
| | Random Search | 64.75% | 0.18M | 0.8264 | 6.37× | 0.8803 |
| | | ±2.15% | ±0.06M | ±0.0588 | ±2.68× | ±0.0152 |
| | Ours | 68.45% | 0.23M | 0.7855 | 4.74× | **0.9171** |
| | | ±1.38% | ±0.04M | ±0.0337 | ±0.78× | **±0.0088** |
| CIFAR-10 | | Accuracy | #Params | Ratio | Times | $f(x)$ |
| VGG-19 | Teacher | 93.91% | 20.04M | - | - | - |
| | Random Search | 91.76% | 2.27M | 0.8865 | 10.54× | 0.9628 |
| | | ±0.88% | ±1.03M | ±0.0515 | ±5.83× | ±0.0149 |
| | N2N | 91.64% | 0.98M | 0.9513 | 20.53× | 0.9735 |
| | Ours | 92.27% | 0.81M | 0.9595 | 25.39× | **0.9809** |
| | | ±0.49 % | ±0.17M | ±0.0084 | ±4.85× | **±0.0050** |
| ResNet-18 | Teacher | 95.24% | 11.17M | - | - | - |
| | Random Search | 92.29% | 0.79M | 0.9293 | 14.42× | 0.9641 |
| | | ±0.83% | ±0.14M | ±0.012 | ±2.39× | ±0.0093 |
| | N2N | 91.81% | 1.00M | 0.9099 | 11.10× | 0.9562 |
| | Ours | 92.99% | 0.85M | 0.9239 | 14.44× | **0.9701** |
| | | ±1.03% | ±0.34M | ±0.0302 | ±5.05× | **±0.0070** |
| ResNet-34 | Teacher | 95.57% | 21.28M | - | - | - |
| | Random Search | 92.87% | 1.70M | 0.9199 | 12.59× | 0.9655 |
| | | ±0.40% | ±0.18M | ±0.0084 | ±1.38× | ±0.0046 |
| | N2N | 92.35% | 2.07M | 0.9020 | 10.20× | 0.9570 |
| | Ours | 92.70% | 1.32M | 0.9379 | 17.00× | **0.9660** |
| | | ±0.74% | ±0.35M | ±0.0163 | ±5.11× | **±0.0072** |
| ShuffleNet | Teacher | 90.87% | 0.99M | - | - | - |
| | Random Search | 88.25% | 0.15M | 0.8490 | 7.38× | 0.9471 |
| | | ±0.51% | ±0.05M | ±0.0529 | ±3.24× | ±0.0095 |
| | Ours | 89.36% | 0.10M | 0.8995 | 10.43× | **0.9729** |
| | | ±1.05% | ±0.03M | ±0.0284 | ±2.54× | **±0.0055** |

$K$ different kernels. This means our proposed method evaluates 160 ($20 \times 8$) architectures in total during the search process. Note that when evaluating an architecture during the search process, we only train it for 10 epochs to reduce computation time. So for both RS and our method, we fully train the top 4 architectures among the 160 evaluated architectures and choose the best one as the solution. When learning the kernel function parameters, we randomly sample from the set of the

evaluated architectures with a probability of 0.5 to form the training set for one kernel. The results of N2N are from the original paper Ashok et al. (2018).

The compression results are summarized in Table 1. For a compressed network $x$, 'Ratio' refers to the compression ratio of $x$, which is defined as $\left(1 - \frac{\#\text{params}(x)}{\#\text{params}(x_{\text{teacher}})}\right)$. 'Times' refers to the ratio between the size of the teacher network and the size of the compressed network, *i.e.*, $\frac{\#\text{params}(x_{\text{teacher}})}{\#\text{params}(x)}$. We also show the value of $f(x)$ as an indication of how well each architecture $x$ meets our requirement in terms of both the accuracy and the compression ratio. For 'Random Search' and 'Ours', we run the experiments for three times and report the average results as well as the standard deviation.

We first apply our algorithm to compress three popular network architectures: VGG-19, ResNet-18 and ResNet-34, and use them as the teacher network. We can see that on both CIFAR-10 and CIFAR-100, our proposed method consistently finds architectures that can achieve higher value of $f(x)$ than all baselines. For VGG-19 on CIFAR-100, the architecture found by our algorithm is 8 times smaller than the original teacher network while the accuracy only drops by $2.3\%$. For ResNet-18 on CIFAR-100, the architecture found by our algorithm has a little bit more parameters than that found by RS but has higer accuracy by about $4\%$. For ResNet-34 on CIFAR-100, the architecture found by our proposed method has a higher accuracy as the architecture discovered by RS but only uses about $65\%$ of the number of parameters. Also for ResNet-34 on CIFAR-100, N2N only provides the results of layer removal, denoted as 'N2N - removal'. 'Ours - removal' refers to only considering the layer removal operation in the search space for fair comparison. We can see that 'Ours - removal' also significantly outperforms 'N2N - removal' in terms of both the accuracy and the compression ratio.

ShuffleNet is an extremely computation-efficient human-designed CNN architecture (Zhang et al., 2018). We also have tried to use ShuffleNet as the teacher network and see if we can further optimize this architecture. As shown in Table 1, our search algorithm successfully compresses 'ShuffleNet $1 \times (g = 2)$' by $10.43\times$ and $4.74\times$ on CIFAR-10 and CIFAR-100 respectively and the compressed architectures can still achieve similar accuracy to the original teacher network. Here '$1\times$' indicates the number of channels in the teacher ShuffleNet and '$(g = 2)$' indicates that the number of groups is 2. Readers can refer to Zhang et al. (2018) for more details about the specific configuration.

## 4.2 Comparison to ShuffleNet

We now compare the compressed architectures found by our algorithm to the state-of-the-art manually-designed compact network architecture ShuffleNet. We vary the number of channels and the number of groups in ShuffleNet and compare the compressed architectures found by our proposed method against these different configurations of ShuffleNet. We conduct experiments on CIFAR-100 and the results are summarized in Table 2. For 'Ours' in Table 2, we use the mean results of 3 runs of our method. In Table 2, VGG-19, ResNet-18, ResNet-34 and ShuffleNet refer to the compressed architectures found by our algorithm based on the corresponding teacher network and do *not* refer to the original architecture indicated by the name. The teacher ShuffleNet used in the experiments is 'ShuffleNet $1 \times (g = 2)$' as mentioned above. '$0.5 \times (g = 1)$' and so on in Table 2 refer to different configurations of ShuffleNet and we show the accuracy of these original ShuffleNet in the table. The compressed architectures found based on ResNet-18 and ResNet-34 have a similar number of parameters with ShuffleNet $1.5\times$ but they can all achieve much higher accuracy than ShuffleNet $1.5\times$. The compressed architecture found based on ShuffleNet $1 \times (g = 2)$ can obtain higher accuracy than ShuffleNet $0.5\times$ while using a similar number of parameters.

## 4.3 Kernel Transfer

We now study the transferability of the learned embedding space or the learned kernel. We would like to know to what extent a kernel learned in one setting can be generalized to a new setting. To be more specific about the kernel transfer, we first learn one kernel or multiple kernels in the source setting. Then we maximize the acquisition function within the search space in the target setting and the acquisition function is computed based on the kernel learned in the source setting. The maximizer of the acquisition function is a compressed architecture for the target setting. We evaluate this architecture in the target setting and compare it with the architecture found by applying algorithms directly to the target setting.

Table 2: Comparison to ShuffleNet on CIFAR-100.

|  | Teacher | Accuracy | #Params | Teacher | Accuracy | #Params |
|---|---|---|---|---|---|---|
| Ours | VGG-19 | 71.41% | 2.61M | ResNet-18 | 73.83% | 1.87M |
|  | ShuffleNet | 68.45% | 0.23M | ResNet-34 | 73.68% | 2.36M |
|  | Configuration | Accuracy | #Params | Configuration | Accuracy | #Params |
| ShuffleNet | $0.5 \times (g = 1)$ | 67.71% | 0.26M | $1.5 \times (g = 1)$ | 72.43% | 2.09M |
|  | $0.5 \times (g = 2)$ | 67.54% | 0.27M | $1.5 \times (g = 2)$ | 71.41% | 2.07M |
|  | $0.5 \times (g = 3)$ | 67.23% | 0.27M | $1.5 \times (g = 3)$ | 71.05% | 2.03M |
|  | $0.5 \times (g = 4)$ | 66.83% | 0.27M | $1.5 \times (g = 4)$ | 71.86% | 1.99M |
|  | $0.5 \times (g = 8)$ | 66.74% | 0.31M | $1.5 \times (g = 8)$ | 71.04% | 2.08M |

Table 3: Summary of Kernel Transfer Results.

|  | Method | Accuracy | Ratio | $f(x)$ | Method | Accuracy | Ratio | $f(x)$ |
|---|---|---|---|---|---|---|---|---|
| (a) $\rightarrow$ (b) | $K = 1$ | 93.13% | 0.8717 | 0.9584 | N2N on (b) | 92.35% | 0.9020 | 0.9570 |
|  | $K = 8$ | 92.80% | 0.9627 | 0.9697 | Ours on (b) | 92.70% | 0.9379 | 0.9660 |
| (a) $\rightarrow$ (c) | $K = 1$ | 89.92% | 0.9793 | 0.9571 | N2N on (c) | 91.64% | 0.9513 | 0.9735 |
|  | $K = 8$ | 92.79% | 0.9671 | 0.9870 | Ours on (c) | 92.27% | 0.9595 | 0.9809 |
| (a) $\rightarrow$ (d) | $K = 1$ | 68.77% | 0.9393 | 0.8708 | N2N on (d) | 68.01% | 0.7845 | 0.8242 |
|  | $K = 8$ | 70.93% | 0.8586 | 0.8835 | Ours on (d) | 73.83% | 0.8335 | 0.9123 |

We consider the following four settings: (a) ResNet-18 on CIFAR-10, (b) ResNet-34 on CIFAR-10, (c) VGG-19 on CIFAR-10 and (d) ResNet-18 on CIFAR-100. 'ResNet-18 on CIFAR-10' refers to searching for a compressed architecture with ResNet-18 as the teacher network for the dataset CIFAR-10 and so on. We first run our search algorithm in setting (a) and transfer the learned kernel to setting (b), (c) and (d) respectively to see how much the learned kernel can transfer to a larger teacher network in the same architecture family (this means a larger search space), a different architecture family (this means a totally different search space) or a harder dataset.

We learn $K$ kernels in the source setting (a) and we transfer all the $K$ kernels to the target setting, which will result in $K$ compressed architectures for the target setting. We report the best one among the $K$ architectures. We have tried $K = 1$ and $K = 8$ and the results are shown in Table 3. In all the three transfer scenarios, the learned kernel in the source setting (a) can help find reasonably good architectures in the target setting without actually training the kernel in the target setting, whose performance is better than the architecture found by applying N2N directly to the target setting. These results proves that the learned architecture embedding space or the learned kernel is able to generalize to new settings for architecture search without any additional training.

## 5  CONCLUSION

We address the task of searching for a compressed network architecture by using BO. Our proposed method can find more efficient architectures than all the baselines on CIFAR-10 and CIFAR-100. Our key contribution is the proposed method to learn an embedding space over the domain of network architectures. We also demonstrate that the learned embedding space can be transferred to new settings for architecture search without any training. Possible future directions include extending our method to the general NAS problem to search for desired architectures from the scratch and combining our proposed embedding space with Hernández-Lobato et al. (2016) to identify the Pareto set of the architectures that are both small and accurate.

ACKNOWLEDGMENTS

The authors would like to thank Sibi Venkatesan for insightful discussions and the reviewers for their valuable feedback.

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

# 6 APPENDIX

## 6.1 DEFINITION OF FUNCTION $f$

We now discuss the form of the function $f$. We aim to find a network architecture which contains as few parameters as possible but still can obtain a similar performance to the teacher network. Usually compressing a network leads to the decrease in the performance. So the function $f$ needs to provide a balance between the compression ratio and the performance. In particular, we hope the function $f$ favors architectures of high performance but low compression ratio more than architectures of low performance but high compression ratio. So we adopt the reward function design in N2N (Ashok et al., 2018) for the function $f$. Formally, $f$ is defined as:

$$f(x) = C(x)\,(2 - C(x)) \cdot \frac{A(x)}{A(x_{\text{teacher}})}, \tag{6}$$

where $C(x)$ is the compression ratio of the architecture $x$, $A(x)$ is the validation performance of $x$ and $A(x_{\text{teacher}})$ is the validation performance of the teacher network. The compression ratio $C(x)$ is defined as $C(x) = 1 - \frac{\#\text{params}(x)}{\#\text{params}(x_{\text{teacher}})}$.

Note that for any $x$, to evaluate $f(x)$ we need to train the architecture $x$ on the training data and test on the validation data. This is time-consuming so during the search process, we do not fully train $x$. Instead, we only train $x$ for a few epochs and use the validation performance of the network obtained by early stopping as an approximation for $A(x)$. We also employ the Knowledge Distillation (KD) strategy (Hinton et al., 2015) for faster training as we are given a teacher network. But when we fully train the architecture $x$ to see its true performance, we fine tune it from the weights obtained by early stopping with cross entropy loss without using KD.

## 6.2 REPRESENTATION FOR LAYER CONFIGURATION

We represent the configuration of one layer by a vector of length $(m + 2n + 6)$, where $m$ is the number of types of layers we consider and $n$ is the maximum number of layers in the network. The first $m$ dimensions of the vector are a one-hot vector, indicating the type of the layer. Then the following 6 numbers indicate the value of different attributes of the layer, including the kernel size, stride, padding, group, input channels and output channels of the layer. If one layer does not have any specific attribute, the value of that attribute is simply set to zero.

Table 4: Ablation study for the number of kernels $K$.

| CIFAR-100 | Accuracy | #Params | Ratio | Times | $f(x)$ |
|---|---|---|---|---|---|
| $K = 1$ | 73.42% | 2.68M | 0.8745 | 7.97x | 0.9181 |
| $K = 2$ | 72.51% | 2.14M | 0.8996 | 9.96x | 0.9119 |
| $K = 4$ | 73.70% | 2.47M | 0.8842 | 8.64x | 0.9238 |
| $K = 8$ | 73.45% | 2.12M | 0.9006 | 10.06x | 0.9240 |
| $K = 16$ | 73.38% | 1.81M | 0.9153 | 11.80x | 0.9256 |

The following $2n$ dimensions encode the edge information of the network, if we view the network as a directed acyclic graph with each layer as a node in the graph. In particular, the $2n$ dimensions are composed of two $n$-dim vectors, where one represents the edges incoming to the code and the other one represents the edges outgoing from the node. The nodes in the directed acyclic graph can be topologically sorted, which will give each layer an index. For an edge from node $i$ to $j$, the $(j-i)^{th}$ element in the outgoing vector of node $i$ and the incoming vector of node $j$ will be 1. We are sure that $j$ is larger than $i$ because all the nodes are topologically sorted. With this representation, we can describe the connection information in a complex network architecture.

## 6.3 ABLATION STUDY

**Impact of number of kernels $K$:** We study the impact of the number of kernels $K$. We conduct experiments on CIFAR-100 and use ResNet-34 as the teacher network. We vary the value of $K$ and fix the number of evaluated architectures to 160. The results are summarized in Table 4. We can see that $K = 4, 8, 16$ yield much better results than $K = 1$. Also the performance is not sensitive to $K$ as $K = 4, 8, 16$ yield similar results. In our main experiments, we fix $K = 8$.

**Impact of adding skip connections:** Our search space is defined based on three operations: layer removal, layer shrinkage and adding skip connections. A key difference between our search space and N2N Ashok et al. (2018) is that they only support layer removal and shrinkage do not support adding skip connections. To validate the effectiveness of adding skip connections, we conduct experiments on CIFAR-100 and on three architectures. In Table 5, 'Ours - removal + shrink' refers to the search space without considering adding skip connections and 'Ours' refers to using the full search space. We can see that 'Ours' consistently outperforms 'Ours - removal + shrink' across different teacher networks, proving the effectiveness of adding skip connections.

**Impact of the maximization of the acquisition function:** As mentioned in Section 3.2, we have two choices to maximize the acquisition function $\text{EI}_t(x)$: randomly sampling (RS) and evolutionary algorithm (EA). We conduct the experiments to compare RS and ES to compress ResNet-34 on CIFAR-100. We find that although EA is empirically better than RS in terms of maximizing $\text{EI}_t(x)$, EA is slightly worse than RS in terms of the final search performance as shown in Table 6. For any $\text{EI}_t(x)$, the solution found by EA $x_{\text{EA}}$ may be better than the solution found by RS $x_{\text{RS}}$, *i.e.*, $\text{EI}_t(x_{\text{EA}}) > \text{EI}_t(x_{\text{RS}})$. However, we observe that $f(x_{\text{EA}})$ and $f(x_{\text{RS}})$ are usually similar. We also plot the values of $f(x)$ for the evaluated architectures when using RS and EA to maximize the acquisition function respectively in Figure 1. We can see that the function value of the evaluated architectures grows slightly more stable when using RS to maximize the acquisition function then using EA. Therefore, we choose RS in the following experiments for the sake of simplicity.

## 6.4 COMPARISON TO TPE (BERGSTRA ET AL., 2011)

Neural architecture search can be viewed as an optimization problem in a high-dimensional and discrete space. There are existing optimization methods such as TPE (Bergstra et al., 2011) and SMAC (Hutter et al., 2011) that are proposed to handle such input spaces. To further justify our idea to learn a latent embedding space for the neural architecture domain, we now compare our method to directly using TPE to search for compressed architectures in the original hyperparameter value domain.

TPE (Bergstra et al., 2011) is a hyperparameter optimization algorithm based on a tree of Parzen estimator. In TPE, they use Gaussian mixture models (GMM) to fit the probability density of the

Table 5: Ablation study for adding skip connections.

| CIFAR-100 | | Accuracy | #Params | Ratio | Times | $f(x)$ |
|---|---|---|---|---|---|---|
| ResNet-18 | Ours - removal + shrink | 72.57% | 1.42M | 0.8733 | 8.85× | 0.9062 |
| | | ±0.58% | ±0.52M | ±0.0461 | ±3.97× | ±0.0081 |
| | Ours | 73.83% | 1.87M | 0.8335 | 6.01× | 0.9123 |
| | | ±1.11% | ±0.08M | ±0.0073 | ±0.26× | ±0.0151 |
| ResNet-34 | Ours - removal + shrink | 73.72% | 2.75M | 0.8711 | 8.01× | 0.9205 |
| | | ±1.33% | ±0.55M | ±0.0257 | ±1.70× | ±0.0117 |
| | Ours | 73.68% | 2.36M | 0.8895 | 9.08× | 0.9246 |
| | | ±0.57% | ±0.15M | ±0.0069 | ±0.59× | ±0.0076 |

Table 6: Ablation study for the maximization of the acquisition function.

| CIFAR-100 | Accuracy | #Params | Ratio | Times | $f(x)$ |
|---|---|---|---|---|---|
| RS, $K = 1$ | 73.42% | 2.68M | 0.8745 | 7.97x | 0.9181 |
| RS, $K = 8$ | 73.45% | 2.12M | 0.9006 | 10.06x | 0.9240 |
| EA, $K = 1$ | 71.52% | 1.24M | 0.9420 | 17.23x | 0.9056 |
| EA, $K = 8$ | 72.40% | 2.15M | 0.8990 | 9.90x | 0.9104 |

hyperparameter values, which indicates that they determine the similarity between two architecture configurations based on the Euclidean distance in the original hyperparameter value domain. However, instead of comparing architecture configurations in the original hyperparameter value domain, our method transforms architecture configurations into a learned latent embedding space and compares them in the learned embedding space.

We first do not consider adding skip connections between layers and focus on layer removal and layer shrinkage only, *i.e.*, we search for a compressed architecture by removing and shrinking layers from the given teacher network. Therefore, the hyperparameters we need to tune include for each layer whether we should keep it or not and the shrinkage ratio for each layer. This results in 64 hyperparameters for ResNet-18 and 112 hyperparameters for ResNet-34. We conduct the experiments on CIFAR-100 and the results are summarized in the Table 7. Comparing 'TPE - removal + shrink' and 'Ours - removal + shrink', we can see that our method outperforms TPE and can achieve higher accuracy with a similar size.

Now, we conduct experiments with adding skip connections. Besides the hyperparameters mentioned above, for each pair of layers where the output dimension of one layer is the same as the input dimension of another layer, we tune a hyperparameter representing whether to add a skip connection between them. The results in 529 and 1717 hyperparameters for ResNet-18 and ResNet-34 respectively. In this representation, the original hyperparameter space is extremely high-dimensional and we think it would be difficult to directly optimize in this space. We can see from the table that for ResNet-18, the 'TPE' results are worse than 'TPE - removal + shrink'. We do not show the 'TPE' results for ResNet-34 here because the networks found by TPE have too many skip connections, which makes it very hard to train. The loss of those networks gets diverged easily and do not generate any meaningful results. Based on the results on 'layer removal + layer shrink' only and the results on the full search space, we can see that our method is better than optimizing in the original space especially when the original space is very high-dimensional.

We would like to point out that TPE (Bergstra et al., 2011) and SMAC (Hutter et al., 2011) focus on improving Sequential Model-Based Optimization (SMBO) methods while our novelty is not in the use of Bayesian optimization methods. Our main contribution is the incrementally learning of an embedding to represent the configuration of network architectures such that we can carry out the optimization over the learned space instead of the original domain of the value of configuration parameters. Our method is complementary to TPE (Bergstra et al., 2011) and SMAC (Hutter et al., 2011) and can be combined with them when being applied to NAS.

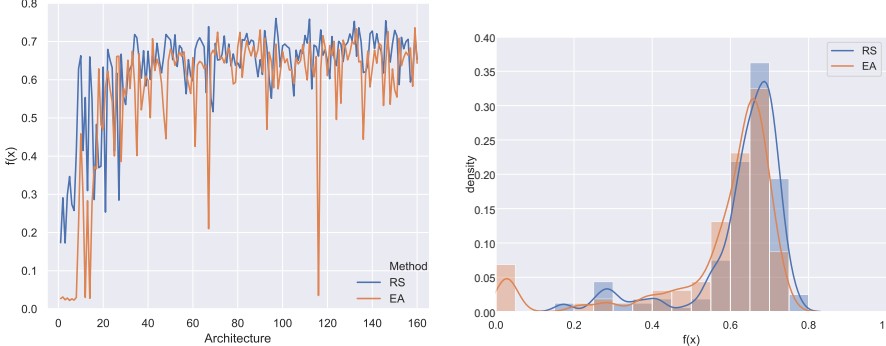

Figure 1: Comparison between random sampling (RS) and evolutionary algorithm (EA) for maximizing the acquisition function. Left: Value of $f(x)$ vs. Index of evaluated architecture. Right: Histogram of values of $f(x)$.

Table 7: Comparison to TPE (Bergstra et al., 2011).

| CIFAR-100 | | Accuracy | #Params | Ratio | Times | $f(x)$ |
|---|---|---|---|---|---|---|
| ResNet-18 | TPE - removal + shrink | 70.60% | 1.30M | 0.8843 | 8.99x | 0.8849 |
| | | $\pm 0.69\%$ | $\pm 0.28M$ | $\pm 0.0249$ | $\pm 2.16x$ | $\pm 0.0111$ |
| | TPE | 65.17% | 1.54M | 0.8625 | 11.82x | 0.8041 |
| | | $\pm 3.14\%$ | $\pm 1.42M$ | $\pm 0.1267$ | $\pm 7.69x$ | $\pm 0.0595$ |
| | Ours - removal + shrink | 72.57% | 1.42M | 0.8733 | 8.85x | 0.9062 |
| | | $\pm 0.58\%$ | $\pm 0.52M$ | $\pm 0.0461$ | $\pm 3.97x$ | $\pm 0.0081$ |
| | Ours | 73.83% | 1.87M | 0.8335 | 6.01x | 0.9123 |
| | | $\pm 1.11\%$ | $\pm 0.08M$ | $\pm 0.0073$ | $\pm 0.26x$ | $\pm 0.0151$ |
| ResNet-34 | TPE - removal + shrink | 72.26% | 2.36M | 0.8893 | 9.24x | 0.9065 |
| | | $\pm 0.83\%$ | $\pm 0.45M$ | $\pm 0.0211$ | $\pm 1.59x$ | $\pm 0.0072$ |
| | Ours - removal + shrink | 73.72% | 2.75M | 0.8711 | 8.01x | 0.9205 |
| | | $\pm 1.33\%$ | $\pm 0.55M$ | $\pm 0.0257$ | $\pm 1.70x$ | $\pm 0.0117$ |
| | Ours | 73.68% | 2.36M | 0.8895 | 9.08x | 0.9246 |
| | | $\pm 0.57\%$ | $\pm 0.15M$ | $\pm 0.0069$ | $\pm 0.59x$ | $\pm 0.0076$ |

## 6.5 RANDOM SAMPLING IN SEARCH SPACE

We need to randomly sample architectures in the search space when optimizing the acquisition function. As mentioned in Section 3.2, we sample the architectures by sampling the operations to manipulate the architecture of the teacher network. During the process, we need to make sure the layers in the network are still compatible with each other in terms of the dimension of the feature map. Therefore, We impose some conditions when we sample the operations in order to maintain the consistency between between layers.

For layer removal, only layers whose input dimension and output dimension are the same are allowed to be removed. For layer shrinkage, we divide layers into groups and for layers in the same group, the number of channels are always shrunken with the same ratio. The layers are grouped according to their input and output dimension. For adding skip connections, only when the output dimension of one layer is the same as the input dimension of another layer, the two layers can be connected. When there are multiple incoming edges for one layer in the computation graph, the outputs of source layers are added up to form the input for that layer. When compressing ShuffleNet, we also slightly modify the original architecture before compression. We insert a $1 \times 1$ convolutional layer before each average pooling layer. This modification increases parameters by about $10\%$ and does not significantly influence the performance of ShuffleNet. Note that the modification only happens when we need to compress ShuffleNet and does not influence the performance of the original ShuffleNet shown in Table 2.

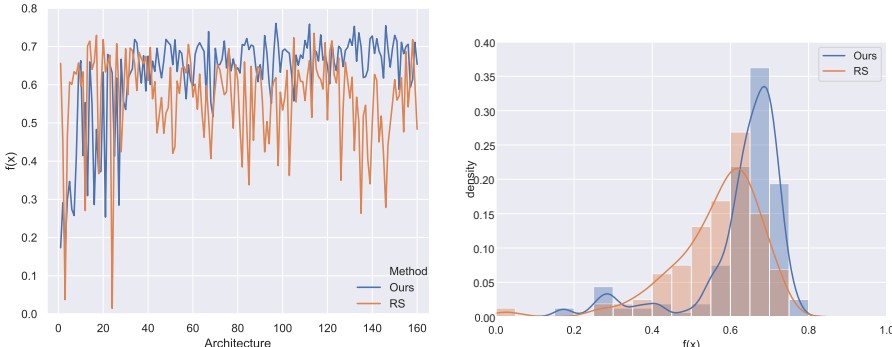

Figure 2: Comparison between our method and random search (RS) baseline. Left: Value of $f(x)$ vs. Index of evaluated architecture. Right: Histogram of values of $f(x)$.

Table 8: Comparison between different objective functions. 'Euclidean' refers to regressing the function value with a Euclidean loss. 'Marginal' refers to maximizing the log marginal likelihood. 'Posterior' is our choice and refers to maximizing the predictive posterior probability.

| CIFAR-100 | | Accuracy | #Params | Ratio | Times | $f(x)$ |
|---|---|---|---|---|---|---|
| VGG-19 | Euclidean | 70.95% | 2.47M | 0.8771 | 9.62× | 0.9453 |
| | | ±1.07% | ±1.26M | ±0.0627 | ±4.55× | ±0.0092 |
| | Marginal | 69.90% | 1.50M | 0.9254 | 16.14× | 0.9422 |
| | | ±0.69% | ±0.68M | ±0.3382 | ±9.22× | ±0.0071 |
| | Posterior | 71.41% | 2.61M | 0.8699 | 7.99× | **0.9518** |
| | | ±0.75% | ±0.61M | ±0.0306 | ±1.99× | **±0.0158** |
| ResNet-18 | Euclidean | 71.67% | 1.62M | 0.856 | 7.07× | 0.8917 |
| | | ±0.67% | ±0.27M | ±0.0243 | ±1.09× | ±0.0137 |
| | Marginal | 72.80% | 1.72M | 0.8467 | 6.57× | 0.9033 |
| | | ±1.11% | ±0.18M | ±0.0160 | ±0.67× | ±0.0094 |
| | Posterior | 73.83% | 1.87M | 0.8335 | 6.01× | **0.9123** |
| | | ±1.11% | ±0.08M | ±0.0073 | ±0.26× | **±0.0151** |
| ResNet-34 | Euclidean | 72.87% | 2.49M | 0.8834 | 8.90× | 0.9127 |
| | | ±1.11% | ±0.60M | ±0.2814 | ±2.04× | ±0.0103 |
| | Marginal | 73.11% | 3.34M | 0.8435 | 6.47× | 0.9059 |
| | | ±0.57% | ±0.48M | ±0.0224 | ±0.89× | ±0.0134 |
| | Posterior | 73.68% | 2.36M | 0.8895 | 9.08× | **0.9246** |
| | | ±0.57% | ±0.15M | ±0.0069 | ±0.59× | **±0.0076** |

## 6.6 ANALYSIS OF RANDOM SEARCH BASELINE

We observe that the random search (RS) baseline which maximizes $f(x)$ with random sampling can achieve very good performance. To analyze RS in more detail, we show the value of $f(x)$ for the 160 architectures evaluated in the search process in Figure 2. The specific setting we choose is ResNet-34 on CIFAR-100. We can see that although RS can sometimes sample good architectures with high $f(x)$ value, it is much more unstable than our method. The function value of the evaluated architectures selected by our method has a strong tendency to grow as we search more steps while RS does not show such trend. Also, from the histogram of values of $f(x)$, we can see that RS has a much lower chance to get architectures with high function values than our method. This is expected since our method leverages the learned architecture embedding or the kernel function to carefully select the architecture for evaluation while RS just randomly samples from the search space. We can conclude that our method is much more efficient than RS.

## 6.7 CHOICE OF THE OBJECTIVE FUNCTION

We discuss the possible choices of the objective function for learning the embedding space in this section. In our experiments, we learn the LSTM weights $\theta$ by maximizing the predictive posterior probability, *i.e.*, minimizing the negative log posterior probability as defined in Eq. 5. There are two other alternative choices for the objective function as suggested by the reviewers. We discuss the two choices and compare them to our choice in the following text.

Intuitively, a meaningful embedding space should be predictive of the function value, *i.e*, the performance of the architecture. Therefore, a reasonable choice of the objective function is to train the LSTM by regressing the function value with a Euclidean loss. Technically, this is done by adding a fully connected layer $FC(\cdot; \theta')$ after the embedding, whose output is the predicted performance of the input architecture. However, directly training the LSTM by regressing the function value does not let us directly evaluate how accurate the posterior distribution characterizes the statistical structure of the function. As mentioned before, the posterior distribution guides the search process by influencing the choice of architectures for evaluation at each step. Therefore, we believe maximizing the predictive posterior probability is a more suitable training objective for our search algorithm than regressing the function value. To validate this, we have tried changing the objective function from Eq. 5 to the squared Euclidean distance between the predicted function value and the true function value: $\frac{1}{|D|} \sum_{i:x_i \in D} (FC(h(x_i; \theta); \theta') - f(x_i))^2$. The results are summarized in Table 8. We observe that maximizing the predictive posterior probability consistently yields better results than the Euclidean loss.

Another possible choice of the objective function is to maximize the log marginal likelihood $\log p\left(f(D) \mid D; \theta\right)$, which is the conventional objective function for kernel learning (Wilson et al., 2016a;b). We do not choose to maximize log marginal likelihood because we empirically find that maximizing the log marginal likelihood yields worse results than maximizing the predictive GP posterior as shown in Table 8. When using the log marginal likelihood, we observe that the loss is numerically unstable due to the log determinant of the covariance matrix in the log marginal likelihood. The training objective usually goes to infinity when the dimension of the covariance matrix is larger than 50, even with smaller learning rates, which may harm the search performance. Therefore, we learn the embedding space by maximizing the predictive GP posterior instead of the log marginal likelihood.

