# OpenReview forum: "Learnable Embedding Space for Efficient Neural Architecture Compression"
_ICLR.cc/2019/Conference_

### Official Review · AnonReviewer2 · 2018-10-23
**A nice paper questioned by the significance of the results**

**Rating:** 6
**Confidence:** 3

**Review:**

Review:

This paper proposes a method for finding optimal architectures for deep neural networks based on a teacher network. The optimal network is found by removing or shrinking layers or adding skip connections. A Bayesian Optimization approach is used by employing a Gaussian Process to guide the search and the acquisition function expected improvement. A special kernel is used in the GP to model the space of network architectures. The method proposed is compared to a random search strategy and a method based on reinforcement learning.

Quality:

	The quality of the paper is high in the sense that it is very well written and contains exhaustive experiments with respect to other related methods

Clarity:

	The paper is well written in general with a few typos, e.g.,

	"The weights of the Bi-LSTM θ, is learned during the search process. The weights θ determines"

Originality:

	The proposed method is not very original in the sense that it is a combination of several known techniques. May be the most original contribution is the proposal of a kernel for network architectures based on recurrent neural networks.

	Another original idea is the use of sampling to avoid the problem of doing kernel over-fitting. Something that can be questioned, however, in this regard is the fact that instead of averaging over kernels the GP prediction to account for uncertainty in the kernel parameters, the authors have suggested to optimize a different acquisition function per each kernel. This can be problematic since for each kernel over-fitting can indeed occur, although the experimental results suggest that this is not happening.

Significance:

	Why N2N does not appear in all the CIRFAR-10 and CIFAR-100 experiments? This may question the significance of the results.

	It also seems that the authors have not repeated the experiments several times since there are no error bars in the results.
	This may also question the significance of the results. An average over several repetitions is needed to account for the randomness in for example the sampling of the network architectures to learn the kernels.

	Besides this, the authors may want to cite this paper

	Hernández-Lobato, D., Hernandez-Lobato, J., Shah, A., & Adams, R. (2016, June). Predictive entropy search for multi-objective Bayesian optimization. In International Conference on Machine Learning (pp. 1492-1501).

	which does multi-objective Bayesian optimization of deep neural networks (the objectives are accuracy and prediction time).

Pros:

	- Well written paper.

	- Simply idea.

	- Extensive experiments.

Cons:

	- The proposed  approach is a combination of well known methods.

	- The significance of the results is in question since the authors do not include error bars in the experiments.

---

> ### Author Response · Authors · 2018-11-15
> **Response to the reviewer's questions; Updated the results of multiple runs; Clarification about the originality**
>
> We thank the reviewer for the feedback and suggestions. We have addressed all the questions here:
>
> *** Response to questions about the performance of N2N: ***
>
> All the numbers of N2N are from their original paper but N2N did not test their method to compress ShuffleNet so we do not have the performance of N2N on ShuffleNet. N2N did not test their method under the setting VGG-19 on CIFAR-100 either. For ResNet-34 on CIFAR-100, N2N only provides results of layer removal (indicated by ‘N2N - removal’ in Table 1 in our paper) so for fair comparison, we compare  ‘N2N - removal’ with ‘Ours - removal’, which refers to only considering the layer removal operation in the search space. ‘Ours - removal’ also significantly outperforms ‘N2N - removal’ in terms of both the accuracy and the compression ratio.
>
>
> *** Response to questions about experiment results: ***
>
> We re-run the experiments for 3 times and update the results in the paper (please check the PDF). In Table 1, we show the mean and standard deviation of the results for ‘Ours’ and ‘Random Search’. We observe that after multiple runs, the average performance of our method also outperforms all the baselines as before.
>
>
> *** Response to questions about the related work: ***
>
> We have updated the paper and added this paper in related work. Also in the conclusion section, we think it’s an interesting future direction to combine their method with our proposed embedding space to identify the Pareto set of the architectures that are both small and accurate. Thanks for suggesting the related work!
>
>
> *** Response to questions about the originality of our work: ***
>
> We would like to emphasize that our key contribution is a novel method that incrementally learns an embedding space for the architecture domain, i.e., a unified representation for the configuration of architectures. The learned embedding space can be used to compare architectures with complex skip connections and multiple branches and we can combine it with any Sequential Model-Based Optimization (SMBO) method (we choose GP based BO algorithms in this work) to search for desired architectures. Based the learned embedding space, we present a framework of searching for compressed network architectures with Bayesian optimization (BO). The learned embedding provides a feature space over which the kernel function of BO is defined. Under this framework, we propose a set of architecture operators for generating architectures for search and a multiple kernel strategy to encourage the search algorithm to explore more diverse architectures.
>
> We demonstrate that our method can significantly outperform various baseline methods, such as random search and N2N (Ashok et al.,2018). The compressed architectures found by our method are also better than the state-of-the-art manually-designed compact architecture ShuffleNet (Zhang et al., 2018). We also demonstrate that the learned embedding space can be transferred to new settings for architecture search, such as a larger teacher network or a teacher network in a different architecture family, without any training.

---

> > ### Comment · AnonReviewer2 · 2018-11-29
> > **Response to author feedback**
> >
> > I have read the authors responses and taken a look and the updated manuscript. They have done a good job. I have hence increased my score accordingly.

---

### Official Review · AnonReviewer1 · 2018-11-02
**interesting idea but the paper needs further work**

**Rating:** 7
**Confidence:** 4

**Review:**

================
Post-Rebuttal
================

I thank the authors for the larger amount of additional work they put into the rebuttal. Since the authors addressed my main concerns, i e. comparison to existing methods,  clarifications of the proposed approach, adding references to related work, I will  increase my score and suggest to accept the paper.




The paper describes a new neural architecture search strategy based on Bayesian optimization to find a compressed version of a teacher network. The main contribution of the paper is to learn an embedding that maps from a discrete encoding of an architecture to a continuous latent vector such that standard Bayesian optimization can be applied.
The new proposed method improves in terms of compressing the teacher network with just a small drop in accuracy upon an existing neural architecture search method based on reinforcement learning and random sampling.


Overall, the paper presents an interesting idea to use Bayesian optimization on high dimensional discrete problems such as neural architecture search. I think a particular strength of this methods is that the embedding is fairly general and can be combined with various recent advances in Bayesian optimization, such as, for instance, multi-fidelity modelling.
It also shows on some compression experiments superior performance to other state-of-the-art methods.

However, in its current state I do not think that the paper is read for acceptance:

- Since the problem is basically just a high dimensional, discrete optimization problem, the paper misses comparison to other existing Bayesian optimization methods such as TPE [1] / SMAC [2] that can also handle these kind of input spaces. Both of these methods have been applied to neural architecture search [3][4] before. Furthermore, since the method is highly related to NASBOT [5], it would be great to also see a comparison to it.

- I assume that in order to learn a good embedding, similar architectures need to be mapped to latent vector that are close in euclidean space, such that the Gaussian process kernel can model any correlation[7]. How do you make sure that the LSTM learns a meaningful embedding space? It is also a bit unclear why the performance f is not used directly instead of p(f|D). Using f instead of p(f|D) would probably also make continual training of the LSTM easier, since function values do not change.

- The experiment section misses some details:
  - Do the tables report mean performances or the performance of single runs? It would also be more convincing if the table contains error bars on the reported numbers.
  - How are the hyperparameters of the Gaussian process treated?

- The related work section misses some references to Lu et al.[6] and Gomez-Bombarelli et al.[7] which are highly related.

- What do you mean with the sentence  "works on BO for NAS can only tune feed-forward structures" in the related work section? There is no reason why other Bayesian optimization should not be able to also optimize recurrent architectures (see for instance Snoek et al.[8]).

- Section 3.3 is a bit confusing and to be honest I do not get the motivation for the usage of multiple kernels. Why do the first architectures biasing the LSTM? Since Bayesian optimization with expected improvement samples around the global optimum, should not later evaluated, well-performing architectures more present in the training dataset for the LSTM?


[1] Algorithms for Hyper-Parameter Optimization
    J. Bergstra and R. Bardenet and Y. Bengio and B. Kegl
    Proceedings of the 25th International Conference on Advances in Neural Information Processing Systems (NIPS'11)

[2] Sequential Model-Based Optimization for General Algorithm Configuration
    F. Hutter and H. Hoos and K. Leyton-Brown
    Proceedings of the Fifth International Conference on Learning and Intelligent Optimization (LION'11)

[3] Towards Automatically-Tuned Neural Networks
    H. Mendoza and A. Klein and M. Feurer and J. Springenberg and F. Hutter
    ICML 2016 AutoML Workshop

[4] Making a Science of Model Search: Hyperparameter Optimization in Hundreds of Dimensions for Vision Architectures
    J. Bergstra and D. Yamins and D. Cox
    Proceedings of the 30th International Conference on Machine Learning (ICML'13)

[5] Neural Architecture Search with Bayesian Optimisation and Optimal Transport
    K. Kandasamy and W. Neiswanger and J. Schneider and B. P{\'{o}}czos and E. Xing
    abs/1802.07191

[6] Structured Variationally Auto-encoded Optimization
    X. Lu and J. Gonzalez and Z. Dai and N. Lawrence
    Proceedings of the 35th International Conference on Machine Learning

[7] Automatic chemical design using a data-driven continuous representation of molecules
    R. Gómez-Bombarelli and J. Wei and D. Duvenaud and J. Hernández-Lobato and B. Sánchez-Lengeling and D. Sheberla and J. Aguilera-Iparraguirre and T. Hirzel. and R. Adams and A. Aspuru-Guzik
    American Chemical Society Central Science

[8] Scalable {B}ayesian Optimization Using Deep Neural Networks
    J. Snoek and O. Rippel and K. Swersky and R. Kiros and N. Satish and N. Sundaram and M. Patwary and Prabhat and R. Adams
    Proceedings of the 32nd International Conference on Machine Learning (ICML'15)

---

> ### Author Response · Authors · 2018-11-15
> **Response to other questions ; Updated results of multiple runs**
>
> Here is our response to other questions:
>
> *** Response to questions about experimental details: ***
> We re-run the experiments for 3 times and update the results in the paper (please check the PDF). In Table 1, we show the mean and standard deviation of the results for ‘Ours’ and ‘Random Search’. We observe that after multiple runs, the average performance of our method also outperforms all the baselines as before.
>
> The mean of the Gaussian process prior is set to zero. The Gaussian noise variance is set to 0.05. The kernel width parameter $\sigma$ (defined in Eq 4) in the RBF kernel is set as $\sigma^2=0.01$.
>
>
> *** Response to questions about related work: ***
>
> Thanks for suggesting the related work. We have updated the paper and added [6] and [7] in the related work section. For your convenience, here is the text about [6] and [7] in the paper: “Our work can also be viewed as carrying out optimization in the latent space of a high dimensional and structured space, which shares a similar idea with previous literature [6][7]. For example, [6] presents a new variational auto-encoder to map kernel combinations produced by a context-free grammar into a continuous and low-dimensional latent space.”
>
> *** Response to “What do you mean with the sentence  "works on BO for NAS can only tune feed-forward structures" in the related work section?”: ***
>
> We are sorry for the confusion of the term ‘feed-forward structures’ in this sentence. We have corrected the sentence to “However, most existing works on BO for NAS only show results on tuning network architectures where the connections between network layers are fixed, i.e., most of them do not optimize how the layers are connected to each other.” For example, [8] tunes the hidden size, the embedding size and other architectural parameters in the language model but it does NOT change how the layers in the model are connected to each other. Our results (Table 5 in Appendix 6.3) show that optimizing how the layers are connected (in this work, by adding skip connections) is crucial to the performance of the compressed network architecture.
>
> The fundamental reason why previous works on BO for NAS do not optimize how the layers are connected is that there lacked a principled way to quantify the similarity between two architectures with complex skip connections, which is addressed by our proposed learnable embedding space. They can benefit our proposed method to be extended to optimize how the layers are connected.
>
> *** Response to questions about the motivation of using multiple kernels: ***
>
> Sorry for the confusion in Sec 3.3. We have edited Sec 3.3 to make the motivation more clear. The main motivation of training multiple kernels is to encourage the search algorithm to explore more diverse architectures. We only evaluate 160 architectures during the whole search process so it is possible the learned kernel is overfitted to the training samples and bias the following sampled architectures for evaluation. To encourage the search algorithm to explore more diverse architectures, we propose the usage of multiple kernels, motivated by the bagging algorithm, which is usually employed to avoid overfitting.
>
> Regarding the statement about the first architecture biasing the LSTM, this statement is invalid in the current context and we have removed it from the paper. This was a conjecture at the early development stage of this work and we mistakenly put it here.

---

> > ### Comment · AnonReviewer1 · 2018-11-19
> > **Question about the treatment of the GP hyperparameters**
> >
> > Does this mean you keep the hyperparameters  of the Gaussian process fixed? Why not adapting them by either optimizing the marginal log-likelihood or marginalizing over them as described in https://papers.nips.cc/paper/4522-practical-bayesian-optimization-of-machine-learning-algorithms.pdf

---

> > > ### Author Response · Authors · 2018-11-21
> > > **Response to questions about the treatment of the GP hyperparameters**
> > >
> > > Yes, the hyperparameters of the GP are fixed in our experiments. We have tried optimizing the kernel width parameter $\sigma$ (defined in Eq 4) and the LSTM weights jointly before but we found that empirically gives worse results. In their experiments [1], the representation of the configuration space is fixed (for example, they represent the architecture configuration with the value of hyperparameters) but in our work, the latent space is learned and keeps being updated during the search process. Optimizing both the GP hyperparameters and the latent space allows more flexibility and may achieve better performance if there are enough training samples for the LSTM. However, in the architecture search scenario, we can only evaluate a few number of architectures, in which case we think fixing the GP hyperparameters and only learning the latent space itself is better.
> > >
> > > [1] Snoek, J., Larochelle, H., & Adams, R. P. (2012). Practical bayesian optimization of machine learning algorithms. In Advances in neural information processing systems (pp. 2951-2959).

---

> ### Author Response · Authors · 2018-11-15
> **Response to Questions about NASBOT [5] and questions about the LSTM training objective**
>
>
> *** Response to the question about NASBOT [5]: ***
>
> Yes, our work is related to NASBOT as mentioned in the related work. Different from our incrementally learned embedding space for the architecture domain, their proposed OTAMANN distance is a *manually defined* distance metric between architectures and can also be used to compare architectures with different topologies. But we find it is non-trivial to integrate OTAMANN distance into our pipeline. Their public implementation is customized to their search space (searching for architectures from the scratch), which is significantly different from our search space (searching for compressed architectures based on a teacher network). Also, to compute OTAMANN distance, one needs to *manually define* a layer label mismatch cost matrix but in their implementation, they treat the residual block as a special layer type while in our work, a residual block is not specially treated but broken down into several layers with skip connections. This makes it hard to integrate OTAMANN distance into our pipeline. We are looking into their code and trying our best for this.
>
> *** Response to “How do you make sure that the LSTM learns a meaningful embedding space?”: ***
>
> The predictive GP posterior guides our choice of the architectures for evaluation at each search step, therefore we learn a meaningful embedding space by updating the LSTM weights θ to maximize \Sum_i log p(f(xi) | f(D \ xi); θ), which is a measurement of how accurate the posterior distribution is. The higher the value of p(f(xi) | f(D \ xi); θ) is, the more accurately the posterior distribution characterizes the statistical structure of the function f and the more the function f is consistent with the GP prior. Thus we define the loss function (Eq 5) based on p(f|D).
>
>
> *** Response to “It is also a bit unclear why the performance f is not used directly instead of p(f|D).”: ***
>
> We agree that a meaningful embedding space should be predictive of the function value (the performance of the architecture). However directly training the LSTM by regressing the function value with a Euclidean loss does not let us directly evaluate how accurate the posterior distribution characterizes the statistical structure of the function. As we have mentioned above, the posterior distribution guides our search process by influencing the choice of architectures for evaluation at each step. Therefore, we believe p(f|D) is a more suitable training objective for our search algorithm than regressing the value of f. To validate this, we have tried changing the objective function from maximizing p(f|D) to regressing the value of f with a Euclidean loss and here are the results:
>
>
> CIFAR-100		                Accuracy	        #Params	        Ratio	                Times	        f(x)
> VGG-19	        Euclidean	70.95%±1.07%	2.47M±1.26M	0.8771±0.0627	9.62x±4.55x	0.9453±0.0092
> 	                Ours	        71.41%±0.75%	2.61M±0.61M	0.8699±0.0306	7.99x±1.99x	0.9518±0.0158
>
> ResNet-18	Euclidean	71.67%±0.67%	1.62M±0.27M	0.8560±0.0243	7.07x±1.09x	0.8917±0.0137
> 	                Ours	       73.83%±1.11%	1.87M±0.08M	0.8335±0.0073	6.01x±0.26x	0.9123±0.0151
>
> ResNet-34	Euclidean	72.87%±1.11%	2.49M±0.60M	0.8834±0.2814	8.90x±2.04x	0.9127±0.0103
> 	                Ours	       73.68%±0.57%	2.36M±0.15M	0.8895±0.0069	9.08x±0.59x	0.9246±0.0076
>
> 'Euclidean' refers to training the LSTM by regressing the value of f with a Euclidean loss. 'Ours' refers to maximizing p(f|D).
>
> We observe that maximizing p(f|D) consistently yields better results than regressing the value of f with a Euclidean loss.

---

> > ### Comment · AnonReviewer1 · 2018-11-19
> > **Response to additional experiments**
> >
> > Thanks for providing the additional experiments. This is very valuable information. Could you add that to the appendix?

---

> > > ### Author Response · Authors · 2018-11-21
> > > **Paper is updated**
> > >
> > > Yes, we have updated the paper and included the additional results in the appendix (see Sec 6.6 and Table 7).

---

> ### Author Response · Authors · 2018-11-15
> **Response to Questions about TPE [1], SMAC [2] and their applications to NAS [3][4]:**
>
> We thank the reviewer for the detailed feedback. Here is our response to questions about  TPE [1], SMAC [2] and their applications to NAS [3][4]:
>
> *** Our key contribution ***
>
> We would like to emphasize that our key contribution is a novel method that incrementally learns an embedding space for the architecture domain, i.e., a unified representation for the configuration of architectures, which includes the number of layers, the type and configuration parameters of each layer and how layers are connected to each other. The learned embedding space can be used to compare architectures with complex skip connections and multiple branches and we can combine it with any Sequential Model-Based Optimization (SMBO) method to search for desired architectures. In this work, we define the kernel function (similarity metric between the configuration of architectures) over this incrementally larned space and apply Bayesian optimization to search for desired architectures. The focus of our work is not the use of Bayesian optimization (or some other SMBO methods) but how the embedding or the representation for the configuration of architectures itself can be learned over time. Other than the Gaussian process regression used in this paper, our method can be combined with more sophisticated SMBO methods such as TPE [1] and SMAC [2]. But this is beyond the focus of this work.
>
>  *** Details about TPE and SMAC ***
>
> TPE [1] is a hyperparameter optimization algorithm based on a tree of Parzen estimator. In TPE [1] and its application to NAS [4],  they use Gaussian mixture models (GMM) to fit the probability density of the hyperparameter values, which indicates that they determine the similarity between two architecture configurations based on the Euclidean distance in the original hyperparameter value domain. However, instead of comparing architecture configurations in the original hyperparameter value domain, we transform architecture configurations into our learned embedding space and compare them in the learned embedding space. Also in [1] and [4], each architectural hyperparameter is optimized independently of others and it is almost certainly the case that the optimal values of some hyperparameters depend on settings of others. This issue can be solved by applying TPE over our learned unified representation for all the configuration parameters.
>
> SMAC [2] is a random-forest-based Bayesian optimization method. In SMAC [2] and its application to NAS [3], they compare two architecture configurations with a combined kernel that is *manually* defined based on the Euclidean distance or the Hamming distance between corresponding configuration parameter values. However, we compare two architecture configurations with an *automatically* learned kernel function defined over a ‘data-driven’ embedding space that is incrementally learned during the optimization. [3] can possibly benefit from our work by replacing their manually defined kernel with our learned kernel function.
>
> *** Our method is complementary to TPE and SMAC ***
>
> Both TPE and SMAC focus on improving SMBO methods while our novelty is not in the use of Bayesian optimization methods. Our main contribution is the incrementally learning of an embedding to represent the configuration of network architectures such that we can carry out the optimization over the learned space instead of the original domain of the value of configuration parameters. Our method is complementary to TPE and SMAC and can be combined with them when being applied to NAS.
>
> *** [3] and [4] do not tune how the layers are connected to each other. ***
>
> Also, TPE [1] and SMAC [2] have been applied to neural architecture search [3][4] before, however the connections between layers in the architectures tuned in [3] and [4] are fixed while we allow the addition of skip connections to optimize how the layers are connected. We believe optimizing how the layers are connected is crucial for the performance of the architecture and we have validated this in the ablation study (Table 5 in Appendix 6.3).

---

> > ### Comment · AnonReviewer1 · 2018-11-19
> > **Response to comparison to existing methods**
> >
> > I do follow the intuition of the paper that, compared to existing methods, the proposed method learns an embedding in order to allow for measuring similarities between architectures in a latent space rather than in the much more complicated original space.
> > It is true that existing BO methods are complementary to the presented approach, however, for me it remains open whether the learned embedding is actually helpful for Bayesian optimization and improves upon methods that only operate in the original input space.
> > I still feel that the paper would be much more convincing, if it contains an experiment that shows that BO in the original space (e.g TPE) is outperformed by Bayesian optimization that uses the latent embedding.

---

> > > ### Author Response · Authors · 2018-11-21
> > > **Will add the results of TPE**
> > >
> > > Thanks for the reply! We agree that the paper would be more convincing if we can compare to applying TPE in the original space. We are running experiments for that and will follow up here once we have the results.

---

> > > ### Author Response · Authors · 2018-11-24
> > > **Comparison to TPE**
> > >
> > > We first do not consider adding skip connections between layers and focus on layer removal and layer shrinkage only, i.e., we search for a compressed architecture by removing and shrinking layers from the given teacher network. Therefore, the hyperparameter we need to tune include for each layer whether we should keep it or not and the shrinkage ratio for each layer. This results in 64 hyperparameters for ResNet-18 and 112 hyperparameters for ResNet-34. The results are summarized in the attached table. Comparing ‘TPE - removal + shrinkage’ and ‘Ours - removal + shrinkage’, we can see that our method outperforms TPE and can achieve higher accuracy with a similar size.
> > >
> > > Now, we conduct experiments with adding skip connections. Besides the hyperparameters mentioned above, for each pair of layers where the output dimension of one layer is the same as the input dimension of another layer, we tune a hyperparameter representing whether to add a connection between them. The results in 529 and 1717 hyperparameters for ResNet-18 and ResNet-34 respectively. In this representation, the original hyperparameter space is extremely high-dimensional and we think it would be difficult to directly optimize in this space. We can see from the table that for ResNet-18, the ‘TPE’ results are worse than ‘TPE - removal + shrink’. We do not show the ‘TPE’ results for ResNet-34 here because the networks found by TPE have too many skip connections, which makes it very hard to train. The loss of those networks gets diverged easily and do not generate any meaningful results.
> > >
> > > Based on the results on ‘layer removal + layer shrink’ only and the results on the full search space, we can conclude that our method is better than optimizing in the original space especially when the original space is very high-dimensional.
> > >
> > >                                                                      Accuracy             #Params            Ratio                    Times               f(x)
> > > CIFAR-100
> > > ResNet-18    TPE - removal + shrink      70.60%±0.69%    1.30M±0.28M    0.8843±0.0249    8.99x±2.16x    0.8849±0.0111
> > >                        TPE                                       65.17%±3.14%    1.54M±1.42M    0.8625±0.1267    11.82x±7.69x   0.8041±0.0595
> > >                        Ours - removal + shrink   72.57%±0.58%    1.42M±0.52M    0.8733±0.0461    8.85x±3.97x    0.9062±0.0081
> > >                        Ours                                     73.83%±1.11%    1.87M±0.08M    0.8335±0.0073    6.01x±0.26x    0.9123±0.0151
> > > ResNet-34    TPE - removal + shrink      72.26%±0.83%    2.36M±0.45M    0.8893±0.0211    9.24x±1.59x    0.9065±0.0072
> > >                       Ours - removal + shrink    73.72%±1.33%    2.75M±0.55M    0.8711±0.0257    8.01x±1.70x    0.9205±0.0117
> > >                       Ours                                     73.68%±0.57%    2.36M±0.15M    0.8895±0.0069    9.08x±0.59x    0.9246±0.0076
> > >
> > >
> > > Caption: ‘TPE - removal + shrink’ and ‘Ours - removal + shrink’ refer to results of TPE and our method when only considering layer removal and layer shrinkage. ‘TPE’ and ‘Ours’ refers to results of TPE and our method when considering the full search space, including layer removal, layer shrinkage and adding skip connections.

---

### Official Review · AnonReviewer3 · 2018-11-07
**interesting idea but...**

**Rating:** 5
**Confidence:** 3

**Review:**

In this work, the authors propose a new strategy to compress a teacher neural network. Briefly, the authors propose using Bayesian optimization (BO) where the accuracy of the networks is modelled using a Gaussian Process function with a squared exponential kernel on continuous neural network (NN) embeddings. Such embeddings are the output of a bidirectional LSTM taking as input the “raw” (discrete) NN representations (when regarded as a covariance function of the “raw” (discrete) NN representations, the kernel is a deep kernel).

The authors apply this framework for model compression. In this application, the search space is the space of networks obtained by sampling reducing operations on a teacher network. In applications to CIFAR-10 and CIFAR-100 the authors show that the accuracies of the compressed network obtained through their method exceeds accuracies obtained through other methods for compression, manually compressed networks and random sampling.

I have the following concerns/questions:

1)	The authors motivate their work in the introduction by discussing the importance of learning a good embedding space over network architectures to “generate a priority ordering of architectures for evaluation”. Within the proposed BO framework, this would require the optimization of the expected improvement in a high-dimensional and discrete space (the space of NN architectures), which “is non-trivial”. In this work, the authors do not try to solve this general problem, but specialize their work to model compression, which has a much lower dimensional search space (space of networks obtained by sampling reducing operations on a teacher network). For this reason, I believe the presentation and motivation of this work is not presented clearly. Specifically, while I agree that the methods and results in this paper can be relevant to the problem of getting NN embeddings for a larger search space, this should be discussed in the conclusion/discussion as future direction, rather than as motivating example. Generally, I think the method should be described in the context of model compression rather than as a general method for neural architecture search (NAS) method (in my understanding, its use for NAS would be unfeasible).

2)	I have been wondering why the authors optimize the kernel parameters by maximizing the predictive GP posterior rather than maximizing the GP log marginal likelihood as in standard GP regression?

3)	The sampling procedure should be explained in greater detail. How many reducing operations are sampled? This would be important to fully understand the random search method the authors consider for comparison in their experiments. I expect that the results from that method will strongly depend on the sampling procedure and different choices should probably be explored for a fair comparison. Do the authors have any comment on this?

---

> ### Author Response · Authors · 2018-11-15
> **Response to the reviewer's questions**
>
> Thanks for the useful feedback! Here is our response:
>
> *** Response to the question about the motivation and the presentation of this paper: ***
>
> We thank the reviewer for the suggestion about the presentation of the paper. We have edited the introduction to motivate our method more in the context of model compression. We also include exploring its application to the general NAS problem as our future work in the conclusion section.
>
>
> *** Response to the question about the using the log marginal likelihood as the objective function: ***
>
> We agree that the log marginal likelihood is the standard objective function in previous works on kernel learning. However, we do not use the log marginal likelihood for the following two reasons:
>
> (1) We empirically find that maximizing the log marginal likelihood yields worse results than maximizing the predictive GP posterior. Here are the results:
>
> CIFAR-100		                        Accuracy		#Params		Ratio		        Times		f(x)
> VGG-19	        Log Marginal	69.90%±0.69%	1.50M±0.68M	0.9254±0.3382	16.14x±9.22x	0.9422±0.0071
> 	                Ours	                71.41%±0.75%	2.61M±0.61M	0.8699±0.0306	7.99x±1.99x	0.9518±0.0158
>
> ResNet-18	Log Marginal	72.80%±1.11%	1.72M±0.18M	0.8467±0.0160	6.57x±0.67x	0.9033±0.0094
> 	                Ours	                73.83%±1.11%	1.87M±0.08M	0.8335±0.0073	6.01x±0.26x	0.9123±0.0151
>
> ResNet-34	Log Marginal	73.11%±0.57%	3.34M±0.48M	0.8435±0.0224	6.47x±0.89x	0.9059±0.0134
> 	                Ours	                73.68%±0.57%	2.36M±0.15M	0.8895±0.0069	9.08x±0.59x	0.9246±0.0076
>
> 'Log Marginal' refers to training the LSTM by maximizing the log marginal likelihood. 'Ours' refers to maximizing p(f|D).
>
> (2) Also, when using the log marginal likelihood, we observe the loss is numerically unstable due to the log determinant of the covariance matrix in the log likelihood. The training objective usually goes to infinity when the dimension of the covariance matrix is larger than 50, even with smaller learning rates, which may harm the search performance.
>
> Therefore, we train the LSTM parameters by maximizing the predictive GP posterior.
>
>
> *** Response to questions about the sampling procedure: ***
>
> Here are the details about how we sample one compressed architecture. This sampling procedure is used in both the ‘Random Search’ baseline and the optimization of the acquisition function in our method.
>
> (1) For layer removal, only layers whose input dimension and output dimension are the same are allowed to be removed. Each removable layer can be removed with probability p_1.  However, if the probability is fixed, the diversity of sampled architectures would be reduced. For example, if we fix p_1 to 0.5, a compressed architecture with over 70% layers removed can hardly be generated. To encourage the diversity of random samples, p_1 is first randomly drawn from the set P_1={0.3, 0.4, 0.5, 0.6, 0.7} at the beginning of generating a new compressed architecture.
>
> (2) For layer shrinkage, we divide layers into groups and for layers in the same group, the number of channels are always shrunken with the same ratio. The layers are grouped according to their input and output dimension. This is to make sure the network is still valid after the layer shrinkage. The shrinkage ratio for each group is drawn from the uniform distribution U(0.0, 1.0).
>
> (3) For adding skip connections, only when the output dimension of one layer is the same as the input dimension of another layer, the two layers can be connected. When there are multiple incoming connections for one layer, the outputs of source layers are added up to form the input for that layer. For each pair of connectable layers, a connection can be added between them with probability p_3. Similar to p_1 in layer removal, p_3 is not fixed but randomly drawn from the set P_3={0.003, 0.005, 0.01, 0.03, 0.05} at the beginning of generating a compressed architecture. Values in P_3 are relatively small, because we found in experiments that adding too many skip connections empirically harm the performance of compressed architectures.
>
> Combining all these three kinds of randomly sampled operations, a compressed architecture is generated from the teacher architecture. We have tried to include more values in the set P_1 and P_3 but that does not yield any improvement in the performance.

---

> > ### Comment · AnonReviewer3 · 2018-12-07
> > **Concerns remain...**
> >
> > Many thanks for the response.
> >
> > Unfortunately, original concerns 2 and 3 remain for me.
> >
> > Specifically, the fact that the authors finally decide to maximize the GP predicitive posterior (and the combination with the multiple Kernel strategy) seems hacky and unprincipled to me. Also the comment "we observe the loss is numerically unstable due to the log determinant of the covariance matrix" makes me worry, did you try to add some jitter to the the diagonal of the covariance (e.g., add 1e-4 * sp.eye(dim_cov))? ()
> >
> > This makes me wonder if the comparison with the random sampling procedure is fair. There are a lot of hyperparameters in the proposed sampling procedure. Did the authors explore whether different values give better performance of the random sampling?
> >
> > Overall, I like the approach taken in this paper and the other reviewers seem to like this work, so I would like to be more supportive. However, at this stage, I do not feel changing my score. Can the authors or the other reviewers make me notice if I am missing anything and/or if (and why) my concerns are seen as unimportant?

---

> > > ### Author Response · Authors · 2018-12-09
> > > **Response to R3’s concerns**
> > >
> > > We thank the reviewer for the feedback. Here are our responses:
> > >
> > > ** Response to the concern about the objective function **
> > >
> > > (1) We agree that maximizing the log marginal likelihood is a principled objective function, but our choice of maximizing the GP predictive posterior p(f(x_i) | f(D \ x_i)) is also reasonable and not a hack. The posterior distribution guides the search process by influencing the choice of architectures for evaluation at each step. The value of p(f(x_i) | f(D \ x_i)) indicates how accurate the posterior distribution characterizes the statistical structure of the function, where f(x_i) is the specific performance value obtained by evaluating the architecture x_i and ‘D \ x_i’ refers to all the evaluated architectures other than x_i. So we believe maximizing p(f(x_i) | f(D \ x_i)) is a suitable training objective for learning the embedding space.
> > >
> > > (2) The key idea of our work is to learn an embedding space for the architecture domain, i.e., a unified representation for the configuration of architectures. The learned embedding space can be used to compare architectures with complex skip connections and multiple branches and we can combine it with any Sequential Model-Based Optimization (SMBO) method to search for desired architectures. While the choice of the objective function itself is important, the idea of mapping an architecture to a latent embedding space is valid no matter what objective function we choose, which is influenced by many factors such as the specific choice of the SMBO method (we choose GP based Bayesian optimization in this work), the principle or the intuition of the objective function and whether the objective function gives good empirical performance or not.
> > >
> > > (3) Yes, we do add a small value on the diagonal of the covariance matrix in our implementation. Here is the code snippet to compute the log determinant in our implementation: “torch.logdet(self.K + self.beta * torch.eye(self.n))”, where K is the covariance matrix, beta is a small positive value and n is the dimension of the matrix.
> > >
> > > We would like to point out that beta here actually refers to the variance of the Gaussian noise. When assuming an additive Gaussian noise with the variance denoted by beta, the formula of the log marginal likelihood naturally contains the term “self.K + self.beta * torch.eye(self.n)” and we do not need to add an extra small value on the diagonal of K.
> > >
> > > ** Response to the concern about the random sampling **
> > >
> > > (1) The comparison between our method and ‘Random Search’ is fair because, in the implementation, our method and ‘Random Search’ use **exactly the same** sampling procedure with the same hyperparameter values to sample architectures. Also, our method and ‘Random Search’ train the same number of architectures in the whole search process.
> > >
> > > The difference between our method and ‘Random Search’ is that ‘Random Search’ randomly samples architectures and train them to get the performance while our method carefully selects the architecture for evaluation by maximizing the acquisition function at each architecture search step. The way we maximize the acquisition function is to randomly sample a set of architectures, evaluate their acquisition function values and choose the architecture with the highest acquisition function value. The randomly sampling procedure used in maximizing the acquisition function in our method is **exactly the same** as the one used in ‘Random Search’. Note that the evaluation of acquisition value for one architecture is super fast, which only involves forwarding the architecture configuration parameters to the LSTM and does not involve any training of this architecture.
> > >
> > > (2) Yes, we have explored different hyperparameter values used in the random sampling procedure, but did not notice an improvement in the performance for either our method or ‘Random Search’.
> > >
> > > (3) Random Search is not the only baseline we have. We have compared our method to N2N (a reinforcement learning based method, see Table 1) and the state-of-the-art manually designed compact architecture ShuffleNet (see Table 2) and have demonstrated superior performance.

---

### Author Response · Authors · 2018-09-28
**Typos in the Paper**

In the right part of Table 2, 'Architecture Teacher #Params' should be 'Teacher Accuracy #Params' and 'Congiguration Teacher #Params' should be 'Configuration Accuracy #Params'.

---

### Public Comment · (anonymous) · 2018-12-17
**Please refer to "Neural Architecture Optimization"**

Dear the authors,

Please refer to the NeurIPS'18 paper: "Neural Architecture Optimization", which maps the NN architectures into their embeddings. It seems a lack of gental discussion (not simple reference) with such a related work is not sound.

Best

---

> ### Author Response · Authors · 2018-12-18
> **Discussion of NAO**
>
> Thanks for the valuable comment. This paper “Neural Architecture Optimization” (NAO) was publicly available on Arxiv at the end of August 2018, about one month before the submission deadline for ICLR. We will add a discussion of NAO in the related work section in the final version of this paper.
>
> NAO and our work share the idea of mapping network architectures into a latent embedding space and carrying out the optimization in this learned embedding space. We are happy to see the idea of mapping network architectures into a latent embedding space has a bigger impact than what’s stated in our paper.
>
> But NAO and our work have fundamentally different motivations for mapping neural network architectures into a continuous space, which further lead to different architecture search frameworks. NAO maps network architectures to a continuous space such that they can perform gradient based optimization to find better architectures. However, our motivation for the embedding space is to make it easy to define a similarity metric (kernel function) between architectures with complex skip connections and multiple branches.
>
> Here is the text from the related work from NAO paper: “However, the effectiveness of GP heavily relies on the choice of covariance functions K(x, x’) which essentially models the similarity between two architectures x and x’. One need to pay more efforts in setting good K(x, x’) in the context of architecture design.” Our work proposes a learnable kernel function K(x, x’) for the architecture domain while NAO does not build upon a Gaussian process and does not touch upon how to define such a kernel function.
>
> The general framework of NAO is gradient based, i.e., NAO selects architectures for evaluation at each architecture search step by using the gradient direction induced by the performance predictor while our search method builds upon Bayesian optimization by defining the kernel function defined over our proposed embedding space.

---

### Public Comment · ~Miao_Zhang1 · 2019-02-08
**Representation of the whole architecture after remove a layer**

Dear authors,

I have thoroughly read this paper, and I found it is very interesting and easy to follow, only one place I can not understand.

You said you use Bi-LSTM to learn the embedding representation, which comes from "the N2N layer removal". My question is, when you remove one layer, do you still use the same length vector to represent the whole architecture? even though several genes are with no meaning? How do you embed the vectors (whole architectures with different number of layers) into same length vectors?

---

> ### Author Response · Authors · 2019-02-08
> **Average pooling ensures the length of the architecture embedding is fixed**
>
> Thanks for your interest in our paper! The embedding for the whole architecture is learned by the Bi-LSTM. After passing the configuration information of each layer into the Bi-LSTM, we gather all the hidden states, apply average pooling to these hidden states and then apply L2 normalization to the pooled vector to obtain the architecture embedding. The average pooling ensures that we obtain a fixed length vector for the whole architecture no matter how many layers we have. The length of the architecture embedding equals the dimension of the hidden state of the Bi-LSTM.

---

> > ### Public Comment · ~Miao_Zhang1 · 2019-02-09
> > **number of the hidden state and the maximal number of layers**
> >
> > Really thanks for your reply! So my question becomes that,  does the number of units in Bi-LSTM equal to the maximal number of layers in the architecture?
> >
> > Sincerely!

---

> > > ### Author Response · Authors · 2019-02-09
> > > **Details about Bi-LSTM**
> > >
> > > At each Bi-LSTM step, we pass the configuration information of one layer to the Bi-LSTM. The input dimension is (m+2n+6). Here n is the maximum number of layers in the network. Details about the representation for layer configuration can be found in Appendix, Sec 6.2.
> > >
> > > The number of Bi-LSTM steps is the same as the number of layers in the network. When one layer is removed, we replace it with an identity layer. The configuration of this 'removed layer' will still be passed to the Bi-LSTM, but here the configuration is updated to an identity layer, different from the original layer. This implementation choice makes the number of Bi-LSTM steps fixed to the number of layers in the given teacher network. But we choose to replace a removed layer as an identity layer instead of actually removing it simply because it is easier to implement and is equivalent to actually removing it. We always get a fixed size embedding for the whole architecture no matter how many layers are in the network because of the average pooling.
> > >
> > > In terms of the architecture details of the Bi-LSTM, we use 4 stacked Bi-LSTM cells and the dimension of the hidden state is 64.

---

### Public Comment · (anonymous) · 2019-04-10
**Code for this project**

Will the code for this project be released? It'd be wonderful if it were released.

---

> ### Author Response · Authors · 2019-04-10
> **Code will be released soon**
>
> Yes, the code will be released soon. Stay tuned!

---

> > ### Public Comment · (anonymous) · 2019-04-17
> > **Time Estimate**
> >
> > Do you have an estimate for when it will be released?

---

> > > ### Author Response · Authors · 2019-04-25
> > > **Code Released**
> > >
> > > Code is available here now: https://github.com/Friedrich1006/ESNAC . Thanks for your interest in our work!

---

### Author Response · Authors · 2019-04-25
**Code Released**

Code is available at https://github.com/Friedrich1006/ESNAC .

---

### Meta-Review · Area_Chair1 · 2018-12-15
**Architecture search through Bayesian Optimization**

**Confidence:** 3
**Recommendation:** Accept (Poster)

**Metareview:**

The authors propose a method to learn a neural network architecture which achieves the same accuracy as a reference network, with fewer parameters through Bayesian Optimization. The search is carried out on embeddings of the neural network architecture using a train bi-directional LSTM. The reviewers generally found the work to be clearly written, and well motivated, with thorough experimentation, particularly in the revised version. Given the generally positive reviews from the authors, the AC recommends that the paper be accepted.